# A Hybrid Multi-Target Path Planning Algorithm for Unmanned Cruise Ship in an Unknown Obstacle Environment

**DOI:** 10.3390/s22072429

**Published:** 2022-03-22

**Authors:** Jiabin Yu, Guandong Liu, Jiping Xu, Zhiyao Zhao, Zhihao Chen, Meng Yang, Xiaoyi Wang, Yuting Bai

**Affiliations:** 1School of Artificial Intelligence, Beijing Technology and Business University, Beijing 100048, China; yujiabin@th.btbu.edu.cn (J.Y.); 1804010402@st.btbu.edu.cn (G.L.); zhaozy@btbu.edu.cn (Z.Z.); 2130062048@st.btbu.edu.cn (Z.C.); 2030602069@st.btbu.edu.cn (M.Y.); wangxy@btbu.edu.cn (X.W.); baiyuting@btbu.edu.cn (Y.B.); 2Beijing Laboratory for Intelligent Environmental Protection, Beijing Technology and Business University, Beijing 100048, China; 3State Environmental Protection Key Laboratory of Food Chain Pollution Control, Beijing Technology and Business University, Beijing 100048, China

**Keywords:** unknown obstacle environment, improved D* Lite algorithm, improved grey wolf optimization algorithm, unmanned cruise ship multi-target path planning

## Abstract

To solve the problem of traversal multi-target path planning for an unmanned cruise ship in an unknown obstacle environment of lakes, this study proposed a hybrid multi-target path planning algorithm. The proposed algorithm can be divided into two parts. First, the multi-target path planning problem was transformed into a traveling salesman problem, and an improved Grey Wolf Optimization (GWO) algorithm was used to calculate the multi-target cruise sequence. The improved GWO algorithm optimized the convergence factor by introducing the Beta function, which can improve the convergence speed of the traditional GWO algorithm. Second, based on the planned target sequence, an improved D* Lite algorithm was used to implement the path planning between every two target points in an unknown obstacle environment. The heuristic function in the D* Lite algorithm was improved to reduce the number of expanded nodes, so the search speed was improved, and the planning path was smoothed. The proposed algorithm was verified by experiments and compared with the other four algorithms in both ordinary and complex environments. The experimental results demonstrated the strong applicability and high effectiveness of the proposed method.

## 1. Introduction

In recent years, Unmanned Cruise Ships (UCSs) for water quality sampling have been widely used in the field of water environment protection. Generally, a UCS needs to traverse multiple target points for water sampling, but there are many unknown obstacles that can move freely and dynamically change with the environment in the actual river or lake, so the UCS is required to plan an optimization path traversing multiple sample points in a short time and effectively avoid unknown obstacles to cruise safely. Therefore, multi-target path planning of a UCS in an unknown obstacle environment is of great importance [1].

Since the 1970s, many studies on the path planning problem have been conducted. The path planning methods can be roughly divided into several groups: the grid search methods, such as A* algorithm [2], Depth-First Search (DFS) [3], Breadth-first Search (BFS) [4], and Dijkstra algorithm [5]; the sampling-based methods, such as Probabilistic Roadmap (PRM) [6] and Rapidly Exploring Random Tree (RRT) [7]; heuristic or swarm intelligence algorithms, such as Genetic Algorithm (GA) [8], Ant Colony Optimization (ACO) [9], Particle Swarm Optimization (PSO) [10], and neural network-based algorithms [11]; the potential field methods, such as Artificial Potential Field (APF) [12], optimal-control based method [13], and geometry-based method [14]. The listed algorithms have certain advantages and disadvantages. For instance, the A* algorithm can perform global path planning in a short time, but it cannot be used in an unknown environment. The APF method has the advantages of easy implementation in local path search, but it can easily fall into a local minimum. It should be noted that complex path planning problems can hardly be solved using a single algorithm. Therefore, hybrid algorithms that combine the advantages of different algorithms have been proposed. For instance, the combination of the A* algorithm and the APF method can achieve path planning in unknown regions [15]. The combination of PSO and ACO can avoid the premature phenomenon of the PSO algorithm, and the convergence speed is also improved [16]. The combination of fuzzy logic algorithm and the dynamic window method can achieve unknown obstacle avoidance [17]. The above-mentioned methods mostly focus on single-target path planning, but multi-target path planning is more complex. Meanwhile, there was little research on the path planning in an unknown obstacle environment.

In order to solve the problem of path planning in an unknown obstacle environment, a hybrid path planning algorithm was proposed in this paper. The main contributions of this work can be summarized as follows:

(i) A hybrid algorithm that combines the advantages of the GWO algorithm and D* Lite algorithm, and thus effectively solves the multi-target path planning problem in an unknown obstacle environment, was proposed.

(ii) An improved GWO algorithm, which optimizes the convergence factor of the traditional GWO algorithm by introducing the Beta function to improve the convergence speed of the algorithm, was developed.

(iii) An improved D* Lite algorithm was proposed. By improving the algorithm’s heuristic function, the expanded nodes are reduced, and the search speed in an unknown obstacle environment is improved. At the same time, the planning path is smoothed.

The rest of the paper is organized as follows. Section 2 presents the related works. Section 3 provides the preliminaries of the grid method, general GWO algorithm, and D* Lite algorithm and defines the UCS path planning problem. Section 4 presents the proposed hybrid multi-target path planning algorithm. Section 5 presents and discusses the experiment results. Finally, Section 6 concludes the paper.

## 2. Related Works

### 2.1. Multi-Target Path Planning

In recent research, the multi-target path planning problem has been mostly regarded as a Traveling Salesman Problem (TSP). Gan et al. [18] introduced the scout strategy to solve the problems of stagnation behavior and slow the convergence speed of the ACO when it was used for the TSP. By adjusting the evaluation model and population size of the algorithm, the search time of the algorithm was shortened. Guo et al. [19] proposed a hybrid algorithm that combines the immune algorithm with the GA. This algorithm introduced a dynamic mutation operator and cross-deletion strategy to solve the TSP, and it can improve the convergence speed and accuracy of the immune algorithm. Liu et al. [20] proposed a Chaos Multi-population Particle Swarm Optimization (CMPSO) algorithm for ship path planning. This algorithm adopts a multi-population strategy to obtain a more accurate global optimal value. Considering the nonlinear dynamics of a vehicle and the dynamic constraint of the TSP, Jang et al. [21] proposed a sampling-based route map algorithm embedded in the route generation process based on optimal control, which represents an easy-to-process way to obtain the closest route planning solution. In recent years, the Grey Wolf Optimization (GWO) algorithm was used to solve the TSP [22]. Compared with the above-mentioned algorithms, such as the GA or PSO, the GWO algorithm has high solution accuracy, simple algorithm operation, only a few parameters to be set, and high robustness. Sopto et al. [23] used the GWO algorithm for numerical optimization in the TSP. The exchange factor and sequence were added in the GWO algorithm. To improve GWO algorithm performance, Xu et al. [24] reconstructed the coding method and target function of the GWO algorithm, and two-opt factor and dynamic, elite strategies were added to obtain a suitable discrete TSP solution. However, the aforementioned GWO algorithms have slow convergence speed and high calculation cost. In addition, solving the TSP provides only an optimization sequence for traversing multiple target points, and for completing multi-target path planning it needs to be combined with other algorithms in the face of unknown obstacles in an actual environment.

### 2.2. Unknown Obstacles Environment

To solve the problem of unknown obstacles in an actual environment, Hossain et al. [25] proposed a method based on bacterial foraging optimization. This method improved the selection criteria of particles, and advanced decision-making strategies were used for particle selection, so the length of the planning path was shortened. To reduce calculation complexity, Hosseininejad et al. [26] used the cuckoo search algorithm for path planning. The dimension of the feature vector was reduced to improve the performance of finding an optimization path, security, and smoothness. Faridi et al. [27] proposed an improved Artificial Bee Colony (ABC) algorithm. The free collocation point and mutation operator were introduced into the ABC algorithm, which effectively improved the obstacle avoidance accuracy in the environment with unknown obstacles. To solve the path planning problem of an Unmanned Aerial Vehicle (UAV), Ma et al. [28] proposed a dynamic augmented multi-target PSO method. The trade-off analysis of different environment targets was implemented to improve the accuracy of the proposed method. To solve the problem of the basic GA in path planning, Liu et al. [29] proposed the concept of visual graphics and safety factors, and the hill climbing algorithm was employed to search better individuals. The dynamic search efficiency of the algorithm was effectively improved. The D* Lite algorithm is an efficient path planning method with the characteristics of flexible change in a dynamic environment. This algorithm has been widely used in an unknown obstacle environment. Huang et al. [30] proposed an improved D* Lite algorithm. This algorithm introduced the concept of a combination of lazy line-of-sight and distance transformation so that the re-planned path can avoid sudden obstacles. To improve the dynamic search efficiency of the D* Lite algorithm, Khalid et al. [31] proposed a concept of predictable obstacles and introduced priority measures to improve the actual search performance of the D* Lite algorithm. However, when an environment map is complex, the above-mentioned D* Lite algorithms have many expanded nodes, so the search efficiency is relatively low, the time cost of the algorithm is high, and the planning path has many inflection points.

Considering the above deficiencies, a hybrid path planning algorithm is proposed in this paper. The proposed algorithm can be divided into two parts. First, based on the environmental map modeling by the grid method, the multi-target planning problem is transformed into a TSP, and the improved GWO algorithm is used to plan a multi-target cruise sequence. Second, based on the obtained sequence, the improved D* Lite algorithm is used for path planning between every two target points. Thus, a circular path that starts from the start point, traverses multiple sampling points, and finally returns to the start point in an unknown obstacle environment is obtained.

## 3. Preliminaries and Problem Formulation

### 3.1. Map Construction

The basic principle of the grid method is that an environment map is divided into independent grid units of the same size according to a certain resolution [32]. In an actual environment, each position is represented by a grid, and a grid has the respective status. In this study, the grid method was used to construct the public water environment map, which denotes a two-dimensional coordinate map. In the grid map, the passable area was represented by a white grid, and the obstacle area was represented by a black grid. When the environment changes, for instance, due to the appearance and disappearance of obstacles or their movement, the grid color corresponding to the changed area also changed accordingly.

### 3.2. GWO Algorithm

The GWO is a heuristic intelligent optimization algorithm proposed by Mirjalili in 2014. The hierarchy of the grey wolf and the search process for prey are simulated in this algorithm. In a group of grey wolves, the level, which is determined by the leadership, is divided into four levels: *α*, *β*, *δ*, and *ω* in a pyramid shape, as shown in Figure 1.

The hierarchy of wolves decreases gradually from top to bottom. Wolves in the upper part have the leadership over wolves in the lower hierarchy, and the wolves in the lower hierarchy generate feedback based on the order of the wolves in the upper hierarchy. By calculating the fitness function of the GWO, the top-three fitness functions are wolves *α*, *β*, and *δ*, and the rest of the wolves are *ω*. According to the position information provided by wolves *α*, *β*, and *δ*, wolves *ω* update their position and move to the prey. According to the related literature [22], the mathematical expressions of the search process are as follows:(1)$$\vec{X}(t+1)={\vec{X}}_{P}(t)-A\cdot \left(C\cdot {\vec{X}}_{P}(t)-\vec{X}(t)\right),$$
where ${\vec{X}}_{}$ is the position vector of a grey wolf, *t* is the current iteration, ${\vec{X}}_{P}$ is the position vector of the prey, *A* = 2*a*∙*r*_1_
*− a* and *C* = 2∙*r*_2_ are coefficients, where *r*_1_ and *r*_2_ are the random numbers in [0, 1], respectively, and *a* is linearly decreased from 2 to 0 over the course of iterations:(2)$$a(t)=2-\frac{2t}{MaxIter},$$
where *t* indicates the current iteration, and *MaxIter* indicates the total number of iterations.

The other wolves update their positions according to the positions of *α*, *β*, and *δ* as follows:(3)$${\vec{X}}_{1}={\vec{X}}_{\alpha }-A_{1}\cdot \left(C_{1}\cdot {\vec{X}}_{\alpha }-\vec{X}\right),$$
(4)$${\vec{X}}_{2}={\vec{X}}_{\beta }-A_{2}\cdot \left(C_{2}\cdot {\vec{X}}_{\beta }-\vec{X}\right),$$
(5)$${\vec{X}}_{3}={\vec{X}}_{\delta }-A_{3}\cdot \left(C_{3}\cdot {\vec{X}}_{\delta }-\vec{X}\right),$$
(6)$$\vec{X}(t+1)=\frac{{\vec{X}}_{1}(t)+{\vec{X}}_{2}(t)+{\vec{X}}_{3}(t)}{3}.$$

The pseudo code of GWO is detailed in literature [22].

### 3.3. D* Lite Algorithm

The D* Lite algorithm is a path planning algorithm proposed by Koenig and Likhachev. It is based on the LPA* algorithm [33]. The D* Lite algorithm represents a heuristic algorithm that can solve the path planning problem in an unknown obstacle environment. Unlike the forward search method of the LPA* algorithm, the D* Lite algorithm uses the reverse search method. Due to the incremental planning, the D* Lite algorithm has a short re-planning time, which makes it suitable for environment maps with unknown obstacles.

Assume that ***S*** denotes the finite set of nodes in a graph, and *Succ* ⊂ *S* denotes the set of successors of a node *s*, *s*∈*S*. The path cost *rhs*(*s*) from the current node *s* to the goal node *s*_goal_ is calculated by:(7)$$rhs(s)=\{\begin{array}{ll} 0 & \mathrm{if}\;s=s_{\mathrm{goal}}\\ \underset{s^{\prime }\in Succ(s)}{\min}(c(s,s^{\prime })+g(s^{\prime })) & \mathrm{otherwise} \end{array},$$
where *c*(*s*, *s*’) denotes the cost of moving from node *s* to node *s*’∈*Succ(s)*; *g*(*s*’) is the actual path cost from the current extension node *s*’ to node *s*_goal_. *rhs*(*s*) is updated earlier than *g(s)*, and all *rhs*(*s*) of the expanded nodes are updated as obstacles appear or disappear, but not all *g(s)* of the nodes need to update with *rhs*(*s*). A detailed description of the above variables can be found in literature [34].

### 3.4. Problem Formulation

The aim of multi-target path planning is to find a path for a UCS that traverses all non-repeating target points. The UCS starts from the start point, traverses all target points, and finally returns to the start point; the path is a circular route. In actual lakes, there are many unknown obstacles, such as ships, moving creatures, floating objects, and various submerged reefs. Therefore, a UCS needs to detect these unknown obstacles and avoid them in time; meanwhile, the planning path needs to be as short as possible to reduce the probability of accidents.

The overall framework of multi-target path planning consists of two modules. The first module aims to obtain a multi-target cruise sequence, which provides a path traversing all non-repeating target points with the criterion of minimum path length. In this process, the unknown obstacles are not considered. Based on the planned target sequence, the second part re-plans the obtained path between every two target points independently using the criterion of the minimum path length and constraint of unknown obstacle avoidance, so a closed path that can guide a UCS traversing all target points safely through an unknown obstacle environment is obtained.

Assume a set of *d* target points, *D* = {1, 2, …, *d*}. The element in *D* represents the serial number of *d* target points. The number of permutations of elements in *D* is *d*!. Let *I* = {*l*_1_, *l*_2_, …, *l_d_*} be a permutation of *D*. A set of all permutations for *d* target points is denoted as *V* = {*I*_1,_
*I*_2_, …, *I_d_*_!_} and an element in *V* represents a cruise sequence for *d* target points. The object is to find a set *I**_i_* (*i*∈{1,2, …, *d*!}) to minimize the following objective function *F*(*I*):(8)$$F(I)=\underset{I_{i}}{\min}\left(\sum_{i=1}^{d-1}L(l_{i},l_{i+1})+L(l_{d},l_{1})\right),i\in \left\{1,2,\ldots d!\right\},$$
where *L*(*l_i_*, *l_j_*) is the Euclidean distance between targets *l_i_* and *l_j_*, and the obtained *I_i_* is the optimized target cruise sequence.

Then, based on *I**_i_*, the path between every two target points is re-planned based on the constraint of unknown obstacle avoidance. This paper considered an area of unknown obstacles, such as ships, moving creatures, floating objects, and various submerged reefs, as a forbidden area that a UCS must avoid. *S_f_* is a set of the paths through the forbidden area. The closed-path *P* is expressed as follows:(9)$$P=\left\{P(l_{1},l_{2}),\ldots ,P(l_{i},l_{i+1})\ldots ,P(l_{d},l_{1})\right\},s.t.P\notin S_{f},$$
where *P*(*l**_i_*, *l_j_*) is the re-planned path between targets *l**_i_* and *l_j_*, and 1 ≤ *i* ≤ *d* − 1.

## 4. Traversal Multi-Target Path Planning

### 4.1. Improved GWO

According to the problem formulation, the considered multi-target path planning problem is similar to the TSP, so it can be transformed into the TSP. The GWO algorithm has been widely used for solving the TSP due to the advantages of few parameters and easy implementation, but it suffers from slow convergence speed. Therefore, this paper first constructed a multi-target coding method according to the optimization path planning requirements of traversing multiple target points. Then, to solve the problem of a slow convergence speed of the GWO algorithm, the convergence factor was improved to increase the convergence speed.

#### 4.1.1. Multi-Target Encoding Construction

The solution range of the traditional GWO algorithm is a two-dimensional continuous space, and the optimization solution can be obtained by determining the value ranges of the target function and independent variables. To solve the TSP of a multi-target cruise sequence, it is necessary to construct a multi-target coding method suitable for the GWO algorithm.

The parameters of the GWO algorithm are as follows: *n* is the number of grey wolves, *d* is the number of target points, and *X_i_* is the position sequence of the *i*th grey wolf traversing *d* target points. The formula of *X_i_* is as follows:(10)$$X_{i}=\left\{x_{i1},x_{i2},\ldots ,x_{id}\right\},i\in \left\{1,2,\ldots ,n\right\},$$
where *x_id_* implies the *i*th grey wolf located at the *d*th target; *n* grey wolves search the optimization sequence in the *d*-dimensional space, and a spatial domain matrix *X* of *n* × *d* can be obtained as follows:(11)$$X=\left[\begin{array}{cccccc} x_{11} & x_{12} & \cdots & x_{1m} & \cdots & x_{1d}\\ x_{21} & x_{22} & \cdots & x_{2m} & \cdots & x_{2d}\\ \vdots & \vdots & \vdots & \vdots & \vdots & \vdots \\ x_{i1} & x_{i2} & \cdots & x_{im} & \cdots & x_{id}\\ \vdots & \vdots & \vdots & \vdots & \vdots & \vdots \\ x_{n1} & x_{n2} & \cdots & x_{nm} & \cdots & x_{nd} \end{array}\right],$$
where *x_im_* implies the *i*th grey wolf located at the *m*th target, and *X_i_* is the *i*th row of matrix *X* and it can be regarded as a sequence to visit *d* target points of the *i*th wolf.

When the number of target points is *d*, the distance between each target can be represented as a *d*-dimensional matrix *P*, which is expressed as:(12)$$P=\left[\begin{array}{cccccc} s_{11} & s_{12} & \ldots & s_{1m} & \cdots & s_{1d}\\ s_{21} & s_{22} & \ldots & s_{2m} & \cdots & s_{2d}\\ \vdots & \vdots & \ddots & \vdots & \ddots & \vdots \\ s_{j1} & s_{j2} & \ldots & s_{jm} & \cdots & s_{jd}\\ \vdots & \vdots & \ddots & \vdots & \ddots & \vdots \\ s_{d1} & s_{d2} & \ldots & s_{dm} & \ldots & s_{dd} \end{array}\right],$$
where *s_jm_* denotes the Euclidean distance between the *j*th and *m*th targets. The diagonal points of the matrix denote the distance from each target to itself, which is why the values of the diagonal points are all zero.

After constructing the distance matrix *P*, it is necessary to construct a fitness function *f*, which represents the shortest sum of distances. The lower the value of the fitness function *f* is, the better the traversal sequence of the target points will be. The fitness function *f* is defined as:(13)$$f=\underset{j}{\min}(\sum_{m=1}^{d}s_{jm}),j\in \left\{1,2,\ldots ,d\right\},$$
where $\sum_{m=1}^{d}s_{jm}$ is the sum of distances between every two target points on the path of the wolf in the *j*th row. When the GWO algorithm reaches the maximum number of iterations, the sequence *X_i_* with the lowest fitness function value is selected as an optimization sequence.

#### 4.1.2. Convergence Factor Improvement

For swarm intelligence algorithms such as the GWO algorithm, it is very important to improve the convergence speed. According to Equations (1) and (2), the convergence factor *a* directly affects the convergence speed. *a* in the traditional GWO algorithm can be expressed in the form of a linear function; it decreases from 2 to 0 as the number of iterations increases. When the maximum number of iterations is reached, the value of *a* is 0, as shown in Figure 2. As the number of iterations increases, *a* is presented as a linear descending line, so the convergence speed of the GWO algorithm is slow. Therefore, this study aims to improve the convergence speed of *a* to increase the convergence speed of the GWO algorithm.

In order to make the initial value of *a* tend to 0, on the basis of the beta function [35], *a* can be expressed as:(14)$$a(t)=\mu_{1}\cdot B(\mu_{2},\frac{t+\lambda_{1}}{t_{\max}})+\lambda_{2},$$
where *t* is the current number of iterations, *t*_max_ is the maximum number of iterations, ***B***(*) is beta function, *μ*_1_ and *μ*_2_ are the position adjustment factors, and *λ*_1_ and *λ*_2_ are the speed adjustment factors.

According to the related literature [36] the specific implementation process of the improved GWO Algorithm is as follows.
**Algorithm:** The Improved GWO Algorithm.Step 1Initialize parametersStep 2Construct matrix ***X*** using Equation (11)Step 3Construct matrix ***P*** and calculate $\sum_{m=1}^{d}s_{jm}$
Step 4Construct fitness function *f* using Equations (12) and (13)Step 5Calculate and update the target sequence using Equations (2)–(6) and (10)–(14), respectivelyStep 6**if** number of iterations < *n*Step 7Repeat Steps 5–7Step 8**else** Output the target sequence

As shown in Figure 2, the improved expression of *a* is a nonlinear decreasing curve. As previously mentioned, *a* tended to be 0 when the maximum number of iterations was reached. The convergence speed of *a* was faster than that in the traditional GWO algorithm. Therefore, the convergence speed of the GWO algorithm can be increased, and the operation time of the algorithm can be reduced. According to Figure 2, the adjustment factor (*λ*_1_ = 1, *λ*_2_ = 0.1) corresponding to the curve with the best convergence is chosen.

### 4.2. Improved D* Lite Algorithm

The traditional D* Lite algorithm is a heuristic algorithm based on the reverse search. The advantage of this algorithm is that it can use previously searched path information to improve the efficiency of the current search, so the calculation burden is reduced. When an environment changes, only the heuristic value and the path cost from the target point to the new start point should be updated, so this algorithm can adapt well to the environment map with unknown obstacles. However, in a complex environment map, due to the rapid increase of the number of expanded nodes, the D* Lite algorithm takes a lot of time, which leads to the low efficiency of the algorithm. Meanwhile, there are many inflection points in the planning path, which is not conducive to the actual cruise. To overcome the two problems, in this study, the heuristic function in the D* Lite algorithm was improved to reduce the number of expanded nodes, so the search efficiency was improved. At the same time, the planning path was smoothed.

#### 4.2.1. Heuristic Function Improvement

The D* Lite algorithm introduces an evaluation function *k(s)*. The node expands in the priority queue with the smallest *k(s)*. The *k(s)* contains two components [*k*_1_*(s)*; *k*_2_*(s)*] as follows:(15)$$\{\begin{array}{l} k_{1}(s)=\min(g(s),rhs(s))+h(s_{\mathrm{start}},s)+k_{m}\\ k_{2}(s)=\min(g(s),rhs(s)) \end{array},$$
where *h*(*s*_start,_
*s*) is the heuristic function that represents the path cost from the start node *s*_start_ to the node *s*. *k(s)* is compared according to a lexicographic ordering. For example, *k(s)* is less than or equal to *k’**(s)*, denoted by *k(s) ≤*
*k’**(s)*, if either *k*_1_*(s) <*
*k*_1_′*(s)* or (*k*_1_*(s)*
*= k*_1_′*(s)* and *k*_2_*(s) ≤*
*k*_2_′*(s)*). *k*_m_ is the superposition of node moving distance and *k*_m_:= *k*_m_ + *h*(*s*_last_, *s*_start_). It is a variable that updates with a change in the environment. It has been shown that the heuristic function *h(s)* directly affects the evaluation function *k(s)*; the mathematical expressions of *h*(*s*_start,_
*s*) are as follows:(16)$$h(s_{\mathrm{start}},s)=\{\begin{array}{ll} 0 & \mathrm{if}\;s=s_{\mathrm{start}}\\ c(s,s^{\prime })+h(s^{\prime },s_{\mathrm{goal}}) & \mathrm{otherwise} \end{array},$$
where *h*(*s*’, *s*_goal_) is the cost function from a node *s*’ to the node *s*_goal_.

In the D* Lite algorithm, the number of expanded nodes directly affects its search efficiency. The evaluation function *k(s)* determines whether a node is expanded. When expanding a node, according to Equation (15), *k*_1_(*s*) of adjacent nodes will be compared preferentially, and *k*_1_(*s*) contains the heuristic function. Therefore, the search efficiency of the algorithm is directly affected by the heuristic function.

Because the heuristic function *h(s)* in the traditional D* Lite algorithm uses the chessboard distance, when expanding nodes near the goal node, it is easy to have multiple nodes with the same value of *k*_1_(*s*). In Figure 3, the black grid represents an obstacle node, and the light grey grid represents the current expanded node. When expanding a node from the goal node, the first step is to initialize the node information. In Step 1, E3 is selected as a goal node to expand its adjacent nodes, and there are three nodes with the same value of *k*_1_(*s*), which are denoted as D2, D3, and D4, and *k*_1_(*s*) values of D2, D3, and D4 are all the smallest value. Thus, the three nodes all need to be expanded gradually. After Step 2, four nodes need to expand, which is represented by the grey grid. It can be observed that there are at least four steps needed to expand the node at layer D. After expanding the three nodes, there are three nodes with the same value of *k*_1_(*s*), which are nodes C1, C2, and C3, and they are all the smallest among all surrounding expanded nodes. Therefore, the same steps are repeated. If many multiple nodes are the same, more nodes need to be expanded, which will increase the calculation time and reduce the search efficiency of the algorithm.

To reduce the number of expanded nodes and increase the search efficiency of the algorithm, according to the related literature [37], an improved heuristic function is proposed. The specific improvements are as follows: horizontal or vertical unit movement cost is defined as one, and the diagonal unit movement cost is defined as $\sqrt{2}$. On this basis, a weight function is added, wherein the greater the value of the weight function is, the farther a node will be from the start node. Therefore, values of *k*_1_*(s)* and *k*_2_*(s)* of multiple nodes are less likely to be equal, the number of expanded nodes is reduced, and the search efficiency is improved. The improved heuristic function *h*’ is defined as follows:(17)$$\left\{ \begin{array}{l} h^{\prime }=\sqrt{2}w\cdot \min(\left|x_{\mathrm{start}}-x\right|,\left|y_{\mathrm{start}}-y\right|)+w\cdot \left|\left|x_{\mathrm{start}}-x\right|-\left|y_{\mathrm{start}}-y\right|\right|\\ w=\frac{\sqrt{(x_{\mathrm{start}}-x{)}^{2}+(y_{\mathrm{start}}-y{)}^{2}}}{\sqrt{(x_{\mathrm{start}}-x_{\mathrm{goal}}{)}^{2}+(y_{\mathrm{start}}-y_{\mathrm{goal}}{)}^{2}}}+1 \end{array}\right.,$$
where *x* and *y* are the horizontal and vertical coordinates of the current node, respectively; *x*_start_ and *y*_start_ are the horizontal and vertical coordinates of the start node, respectively; *x*_goal_ and *y*_goal_ are the horizontal and vertical coordinates of the goal node, respectively; *w* is the weight factor, and its range is [1, 2].

The specific process of the improved D* Lite algorithm is presented in Figure 4. First, the node information is initialized, then the adjacent nodes of node E3 are expanded. In Step 1, the node information is obtained by Equation (17). The node with the smallest value of *k*_1_(*s*) is node D2, so the other two nodes, nodes D3 and D4, are no longer needed to be expanded, and only one step is needed to expand the node at layer D. In Step 2, node D2 is selected for expansion. Based on the expansion result, a node with the smallest value of *k*_1_(*s*) is node C1, and there are no other nodes with the same values. Therefore, after Step 2, the node expansion at layer D is completed and there are only two expansion nodes (grey grid). However, by the traditional D* Lite algorithm in Figure 3, the number of expansion nodes is four, so the improved D* Lite algorithm of the heuristic function reduces two expanded nodes, and therefore has a higher search efficiency. 

The comparison results of the traditional D* Lite algorithm and the proposed D* Lite algorithm were shown in Figure 5, where the black grid represented an obstacle, the grey grid represented the expanded node, and the red grid represented the final path. It was shown that, in the case of the same obstacles, although the planning paths of the two methods were the same, the number of expanded nodes in the improved D* Lite algorithm were obviously less than in the traditional D* Lite algorithm. Thus, the search efficiency was improved, and the calculation time of the algorithm was effectively reduced.

#### 4.2.2. Path Smoothing

When the traditional D* Lite algorithm is used for path planning, there are many inflection points in the planned path. The inflection points will increase the path length, as well as the control difficulty and power consumption of the UCS. Therefore, it is necessary to smooth the planning path by eliminating inflection points. According to the related literature [38], the specific implementation process of path smoothing is as follows.**Algorithm:** Path smoothing.Step 1Label each point on the planning path from one to *n*Step 2Connect points 1 and 2 and check whether the connection passes through the obstaclesStep 3Check until the connection between points 1 and *k* (*k* < *n*) passes through the obstacleStep 4Connect point 1 and (*k* − 1) and replace the previous path from point 1 to (*k* − 1)Step 5Use point (*k* − 1) as a new start point and repeat the above steps until the target is reached

Based on Section 4.2.1 and Section 4.2.2, according to the related literature [33], the specific implementation process of the improved D* Lite Algorithm is as follows.
**Algorithm:** Improved D* Lite algorithm.Step 1Parameter initializationStep 2Expand adjacent nodes from *s*_goal_Step 3Compare current *k*(*s*) values and select the node with the smallest *k*(*s*) as the next expanded nodeStep 4Expand the nodes constantly until reach *s*_start_
Step 5Calculate the values of *rhs*(*s*) and move to the node with the smallest *rhs*(*s*) Step 6**If** the surrounding environment has changedStep 7update adjacent nodes and return to Step 2Step 8**else** the current node is the new start node *s’*_start_Step 9**If** node *s’*_start_ is node *s*_goal_
Step 10Perform path smoothingStep 11**else** return to Step 2Step 12Complete path planning between every two target points

### 4.3. Algorithm Overview

The specific process of the proposed hybrid multi-target path planning algorithm is as follows. First, a UAV is used to obtain an image of the water environment, which is then transformed into a two-dimensional coordinate map by the grid method, and the coordinates of multiple target points are set on the map. Second, the proposed improved GWO algorithm is used to obtain the sequence of multiple target points, and the serial number of target points is marked on the grid map according to the planned sequence. Third, the improved D* Lite algorithm is used to calculate a path between every two target points in the grid map, and the planning path is smoothed. Finally, a closed path, which starts from the start point, traverses multiple target points, and returns to the start point, is obtained.
**Algorithm:** The proposed multi-target hybrid path planning algorithm.Step 1Parameter initializationStep 2Introduce improved convergence factorStep 3Calculate fitness functionStep 4Determine target sequence of *α*, *β*, *δ*, and *ω*Step 5**If** the maximum number of iterations is reachedStep 6Output the target sequenceStep 7**else** Update adjacent nodes and return to Step 2Step 8Perform path planning between two target points by the improved D* Lite algorithm Step 9**If** the surrounding environment has not changedStep 10**If** the new start point *s’*_start_ is the target pointStep 11   **If** it is the final target pointStep 12    return to the start pointStep 13   **else** return to Step 8Step 14  **else** return to Step 8Step 15**else** return to Step 8Step 16Perform path smoothingStep 17Complete multi-target path planning

## 5. Simulation Experiments

To verify the advantages of the proposed hybrid algorithm, several simulation experiments were performed. Windows 10 was used as an operating system and MATLAB R2017b as a simulation tool. The hardware platform was an Intel Core E5-2620 V3 processor with a frequency of 2.4 GHz and a memory of 32 GB. The simulation environment map was created based on the public water area of Douhu, southwest of Hongze Lake, located in Jiangsu Province. The simulation environment map was created by grid method [32]. The simulation experiments were divided into two groups, experiments in ordinary and complex environments, mainly based on the grid map size and the number of target points.

### 5.1. Simulations in Ordinary Environments

A 50 × 50 grid map model was created in an ordinary environment. The scale of the map was one, and a grid length corresponded to 10 m in the actual environment. The UCS used a laser detector that could detect the surrounding obstacles in time, and the detection distance was 5 × 5 grid. The speed of the UCS was 10 m/s. The target coordinates were set randomly; four sets of the target coordinates were randomly chosen, and they were denoted as Cases 1–4 in Table 1. The number of target points was 10, and the serial number of the start point was marked as one. In the map, the randomly-distributed obstacle density was 10%, and the settings were as follows: each grid had two states, white or black, where the white grid represented the passable area, and the black grid represented the obstacle area. The state of each grid had a certain probability of changing. In this experiment, the change probability of the grid state was set to 3%, so when the UCS moved to a grid, the probability of a white grid changing to a black grid was 3%, and the probability of a black grid changing to a white grid was also 3%.

To verify the performance of the proposed hybrid algorithm, comparative experiments with the other four algorithms were performed. The comparison algorithms were denoted as Algorithms 1–4. Algorithm 1 used the ACO to determine the cruise sequence of target points. According to the planned sequence, the ACO was used again to achieve path planning between two target points. Similarly, Algorithm 2 used the GA to complete the multi-target path planning. Algorithm 3 used the CMPSO to complete the multi-target path planning. Algorithm 4 used the traditional GWO algorithm–traditional D* Lite algorithm to complete the multi-target path planning. The proposed hybrid algorithm used the improved GWO algorithm to determine the cruise sequence of target points, and the improved D* Lite algorithm was used for the single-target path planning. According to [39], the parameters of the ACO in Algorithm 1 were set as follows: the information priming factor was set to *α* = 1, the expected heuristic factor was set to *β =* 5, the volatilization factor was set to *ρ =* 0.1, the pheromone intensity was set to *Q =* 1, the number of ants was set to *m* = 200, and the maximum number of iterations was set to *k* = 100. According to [40], the parameters of the GA in Algorithm 2 were set as follows: the maximum number of evolution times was set to *max* = 50, the crossing probability was set to *pc* = 0.8, the mutation probability was set to *pm* = 0.2, the path length proportion was set to *a* = 1, and the path smoothness proportion was set to *b* = 7. According to [20], the parameters of the CMPSO in Algorithm 3 were set as follows: the inertia weight was set to *ω* = 0.4, and the learning factor was set to *c*_1_ = *c*_2_ = 2. The simulation parameters of Algorithm 4 and the proposed hybrid algorithm were presented in Table 2. The adjustment factors *λ*_1_ and *λ*_2_ were chosen according to the better convergence curve through a series of experiments, as shown in Figure 3.

The five algorithms were used for comparison experiments in Case 1, and when the UCS returned to the start point, the obtained simulation results were shown in Figure 6, Figure 7, Figure 8, Figure 9 and Figure 10. All five algorithms could complete the multi-target path planning; the planned target sequence was the same, but the planning path was different. The red numbers in Figure 6, Figure 7, Figure 8, Figure 9, Figure 10, Figure 11, Figure 12, Figure 13, Figure 14 and Figure 15 represent the traversal sequence between every two target points. The detection area in path planning in Figure 10 was obviously smaller than that in Figure 9, which indicated the effect of the proposed algorithm.

To avoid contingency and verify the robustness of the proposed algorithm, 50 simulation experiments were performed for Case 1 using the five algorithms in Table 3. According to the related literature [41], the *t*-test was used to verify whether the proposed hybrid algorithm significantly improved the path planning performance. The data of the proposed hybrid algorithm were taken as the total samples, the other four algorithms were used as test samples, and the significance difference was 0.05. When the t-test value was less than 0.05, it meant that the test performance of the algorithm was significantly different from that of the proposed algorithm, and the significant improvement was indicated by ‘+’. In addition to the planning time and length, the number of inflection points was also used as an evaluation metric of algorithm performance since it can reflect the smoothness of the path. Too many inflection points would increase the UCS’s energy consumption and reduce its safety. The statistical analysis of the results are shown in Table 3, where it can be seen that the proposed hybrid algorithm always found an optimal solution and converged to the stable state, and its performance was better than those of the other four comparative algorithms.

In addition, to verify the efficiency and generalizability of the proposed algorithm, the other three cases were simulated using different target coordinates. The algorithm performance comparison was given in Table 4, where it can be seen that in terms of planning time, planning length, and the number of inflection points, the proposed hybrid algorithm performed better than the other four comparative algorithms. Consequently, the proposed hybrid algorithm had stronger applicability and higher performance than the other four algorithms.

### 5.2. Simulations in Complex Environments

In complex environments, a 100 × 100 grid map model was created. The number of target points was 20, and the serial number of the start point was marked as one. In this map, the randomly-distributed obstacle density was 12%. The target coordinates were set randomly; four sets of the target coordinates were randomly selected, and they are denoted as Cases 1–4 in Table 5.

The parameters of the five algorithms were the same as in the previous experiments. The experimental results were shown in Figure 11, Figure 12, Figure 13, Figure 14 and Figure 15. In complex environments, the five algorithms could complete the multi-target path planning, but the planned sequence of the target points presented in Figure 11, Figure 12 and Figure 13 were different from those in Figure 14 and Figure 15. Obviously, the grey detection area in Figure 14 was larger than that in Figure 15 and the paths in Figure 15 were smoother than those of the other four algorithms.

Fifty simulation experiments of the five algorithms in Case 1 were performed in complex environments. The statistical analysis of the experimental results was shown in Table 6. The results in Table 6 further validated the robustness of the proposed algorithm in complex environments. In the *t*-test, the proposed algorithm performed significantly better than the other four hybrid algorithms. Compared with the ordinary environments, the proposed hybrid algorithm showed more obvious advantages in complex environments. The other three cases were also simulated using different target coordinates, and the results were presented in Table 7. Similarly, the performance of the proposed hybrid algorithm was better than of the other four comparison algorithms.

### 5.3. Performance Testing of the Proposed Algorithm

To investigate the performance of the proposed algorithm further, according to [42], four benchmark test functions were chosen for comparison experiments, and they have been given in Table 8.

The four benchmark test functions were solved by the Improved Grey wolf optimization algorithm (IGWO) and compared with the numerical calculation results of the Ant colony optimization, Genetic algorithm, Chaos multi-population particle swarm optimization, and Grey wolf optimization algorithms. The parameters of these algorithms were the same as in the above-mentioned experiments. The experiments with each of the functions were run independently 30 times, and the mean and standard deviation of the algorithms were recorded. The comparison results of five algorithms for the optimization of *f*_1_–*f*_4_ were shown in Table 9. For function *f*_3_, the results of the proposed Improved Grey wolf optimization algorithm tended to converge to the theoretical optimal value of zero. For functions *f*_1_, *f*_2_, and *f*_4_, the experimental results of the Improved Grey wolf optimization algorithm were also close to the optimal value. In terms of solution accuracy (mean value) and robustness (standard deviation), the performance of the proposed Improved Grey wolf optimization algorithm was significantly superior to those of the other four algorithms.

The iterative average convergence curves of the five algorithms for the four functions are shown in Figure 16, where it can be seen that the IGWO had a faster convergence rate than the other algorithms.

## 6. Conclusions

In this paper, a hybrid algorithm based on the improved GWO-D* Lite algorithm was proposed. First, the multi-target planning problem was transformed into a TSP, and the improved GWO algorithm was used to plan a multi-target cruise sequence. Second, based on the obtained sequence, the improved D* Lite algorithm was used for path planning between every two target points. The simulation verification of the proposed algorithm was conducted in both ordinary and complex environments. The comparative simulation experiments with the other four algorithms were also implemented. To avoid contingency and verify the efficiency and generalizability of the proposed algorithm, a detailed statistical analysis based on multiple experiments and a performance comparison with different target coordinates were performed. The simulation results show that, in terms of planning time, planning distance and number of inflection points, the proposed algorithm has obvious advantages over the other four algorithms. Meanwhile, the results of four standard test functions show that the proposed algorithm has strong optimization ability. The proposed algorithm can provide important guidance for multi-target path planning of UCS in an unknown obstacle environment and promote the development of intelligent technology of UCS.

In future research, the proposed algorithm could be improved in two aspects. First, a number of the proposed algorithm’s parameters were obtained by experience, so they were influenced by subjective factors. Therefore, these parameters could be optimized by certain methods. Second, the grid map was a 2D model, so it had limitations in reflecting an actual environment. In future research, a 3D map model could be built for an actual environment, which can more accurately reflect the actual shape of a UCS and obstacles. In addition, the currents in the lake, the health status, dynamic behavior, and motion limitation of a UCS could also be constraints of the path optimization problem.

## Figures and Tables

**Figure 1 sensors-22-02429-f001:**
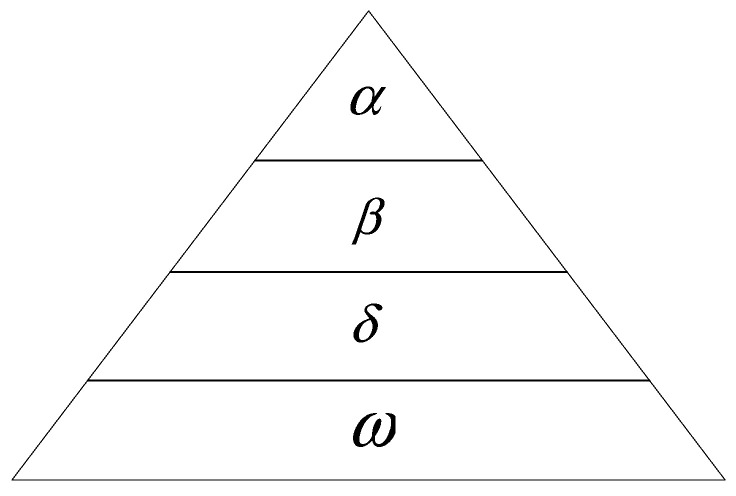
Wolf social hierarchy pyramid.

**Figure 2 sensors-22-02429-f002:**
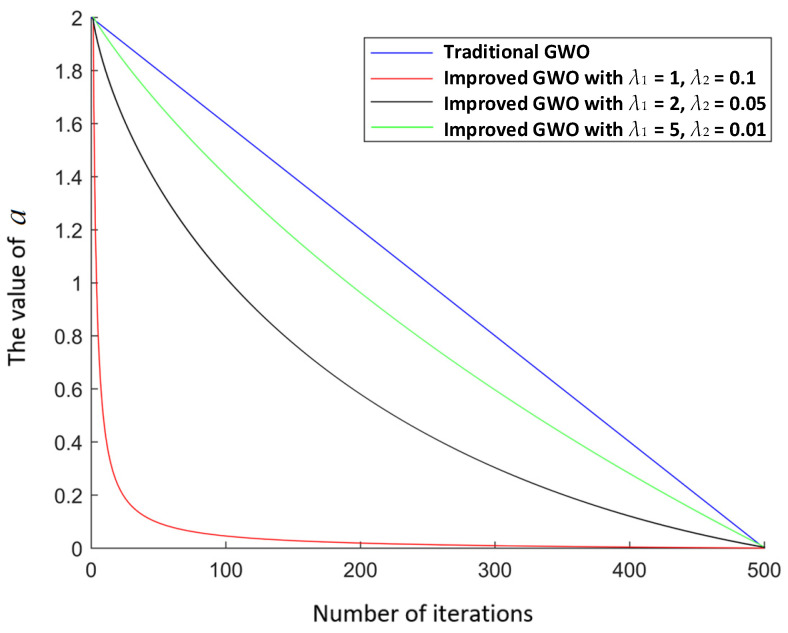
The convergence comparison chart of *a* with different *λ*_1_ and *λ*_2_.

**Figure 3 sensors-22-02429-f003:**
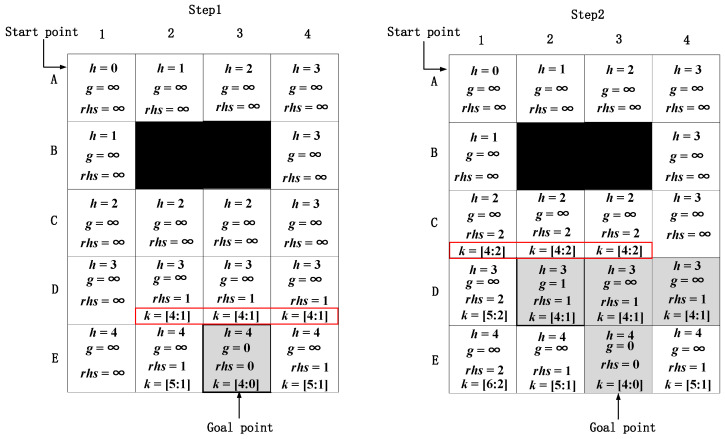
The specific process of the traditional D* Lite algorithm.

**Figure 4 sensors-22-02429-f004:**
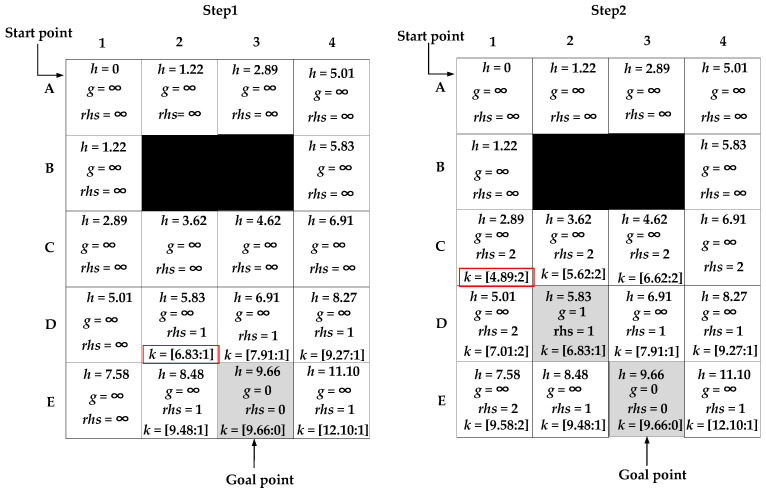
The specific process of the improved D* Lite algorithm.

**Figure 5 sensors-22-02429-f005:**
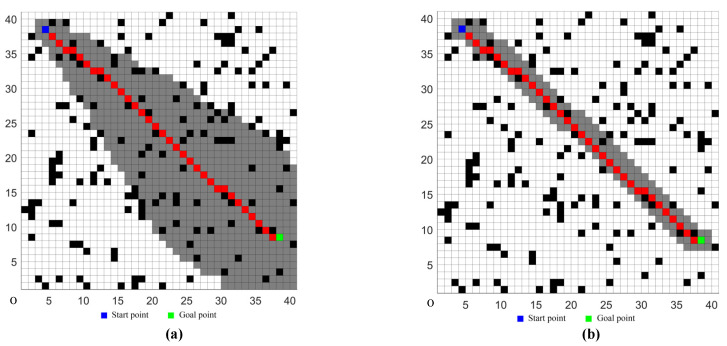
(**a**) The traditional D* Lite algorithm. (**b**) The improved D* Lite algorithm.

**Figure 6 sensors-22-02429-f006:**
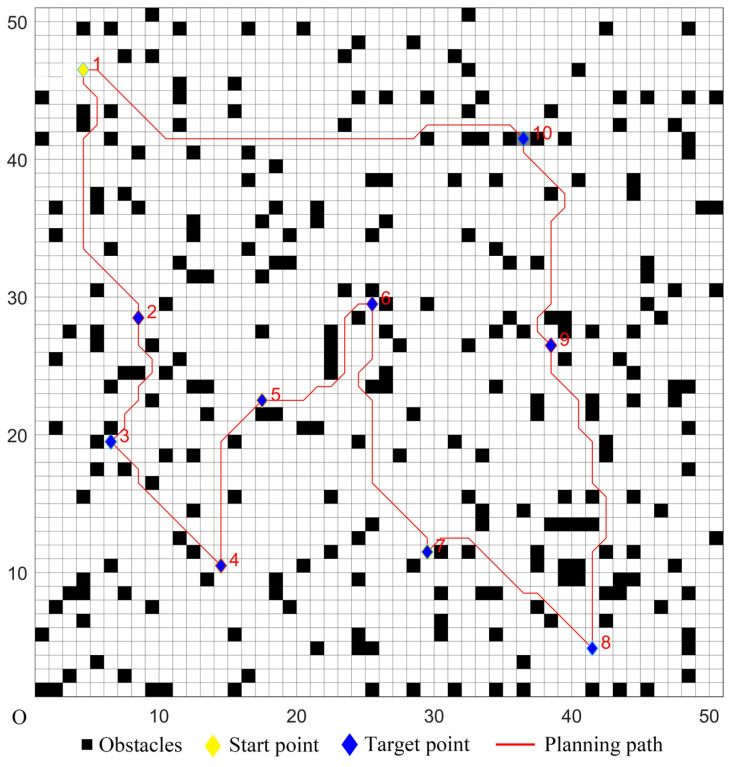
Multi-target path planning results of Algorithm 1 in ordinary environments.

**Figure 7 sensors-22-02429-f007:**
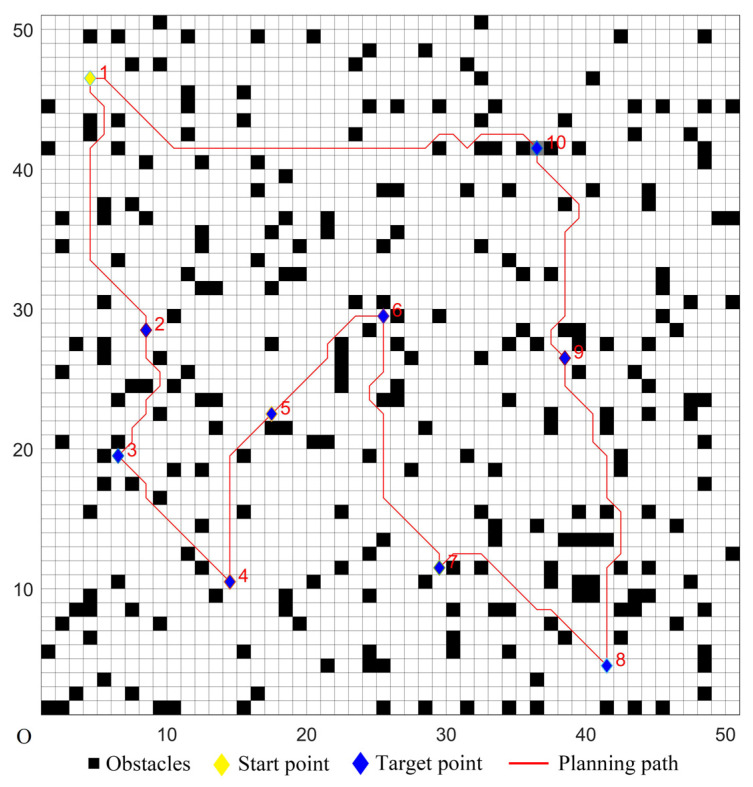
Multi-target path planning results of Algorithm 2 in ordinary environments.

**Figure 8 sensors-22-02429-f008:**
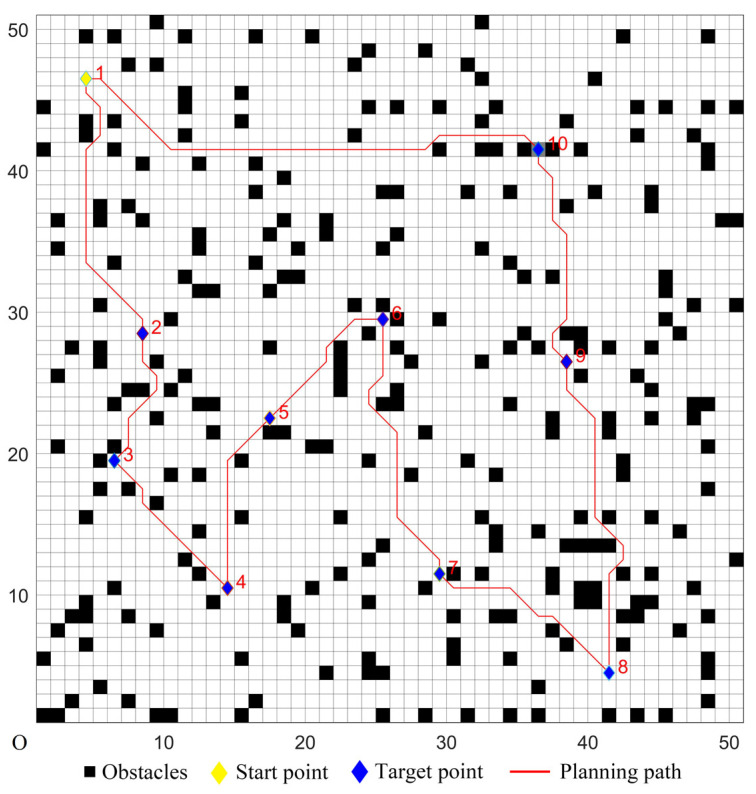
Multi-target path planning results of Algorithm 3 in ordinary environments.

**Figure 9 sensors-22-02429-f009:**
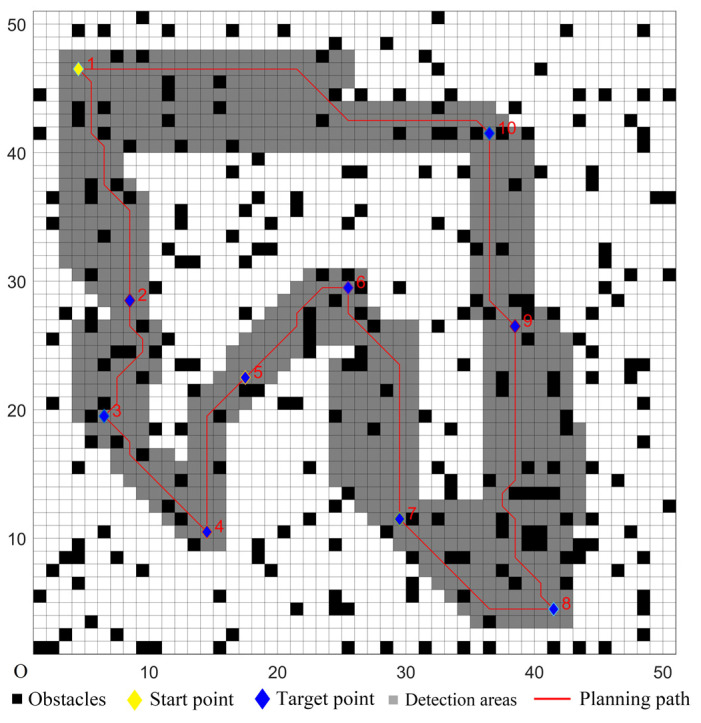
Multi-target path planning results of Algorithm 4 in ordinary environments.

**Figure 10 sensors-22-02429-f010:**
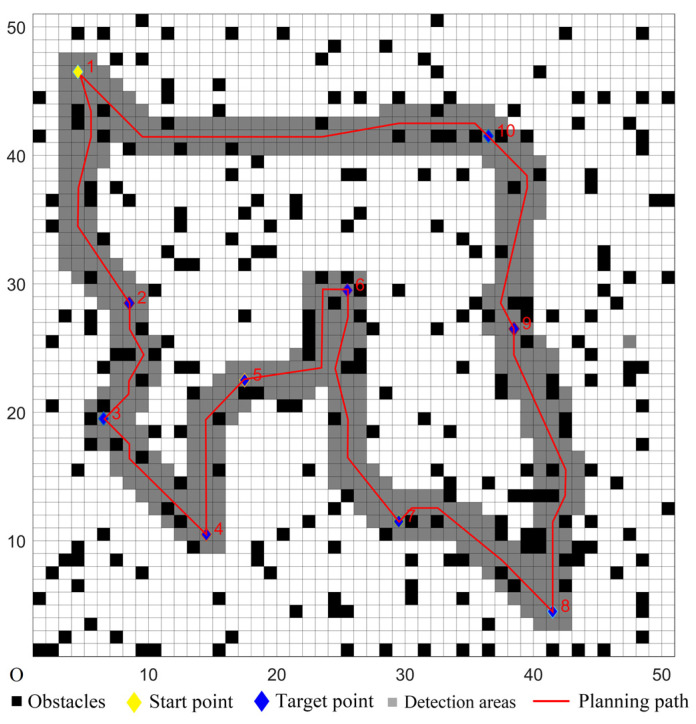
Multi-target path planning results of the proposed hybrid algorithm in ordinary environments.

**Figure 11 sensors-22-02429-f011:**
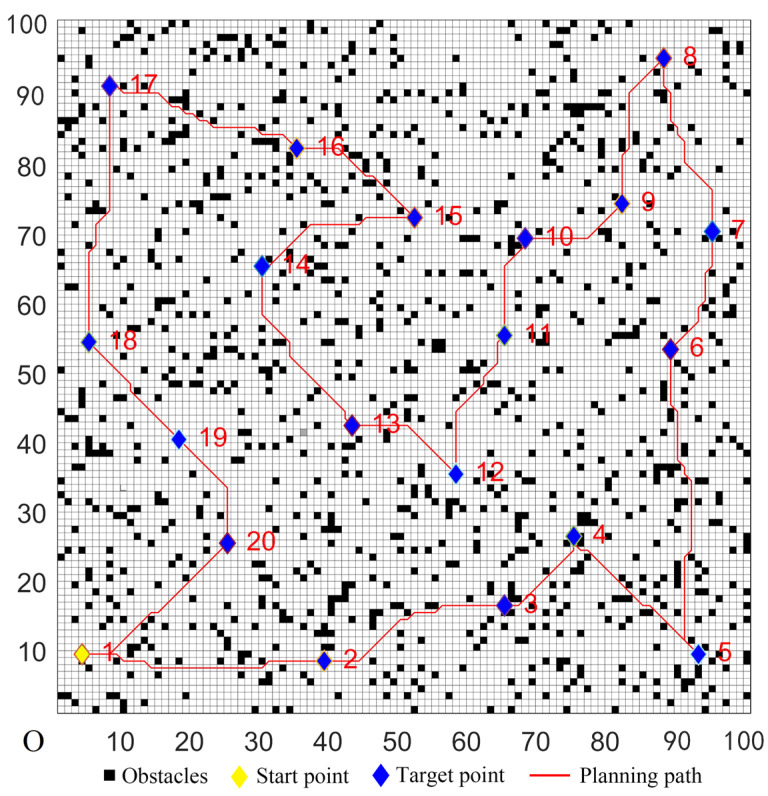
Multi-target path planning results of Algorithm 1 in complex environments.

**Figure 12 sensors-22-02429-f012:**
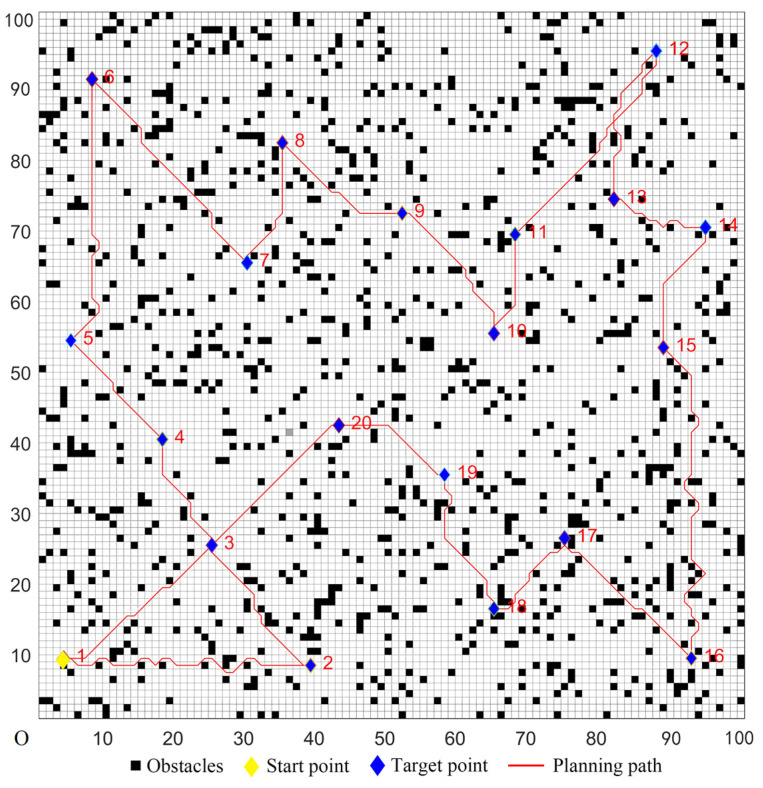
Multi-target path planning results of Algorithm 2 in complex environments.

**Figure 13 sensors-22-02429-f013:**
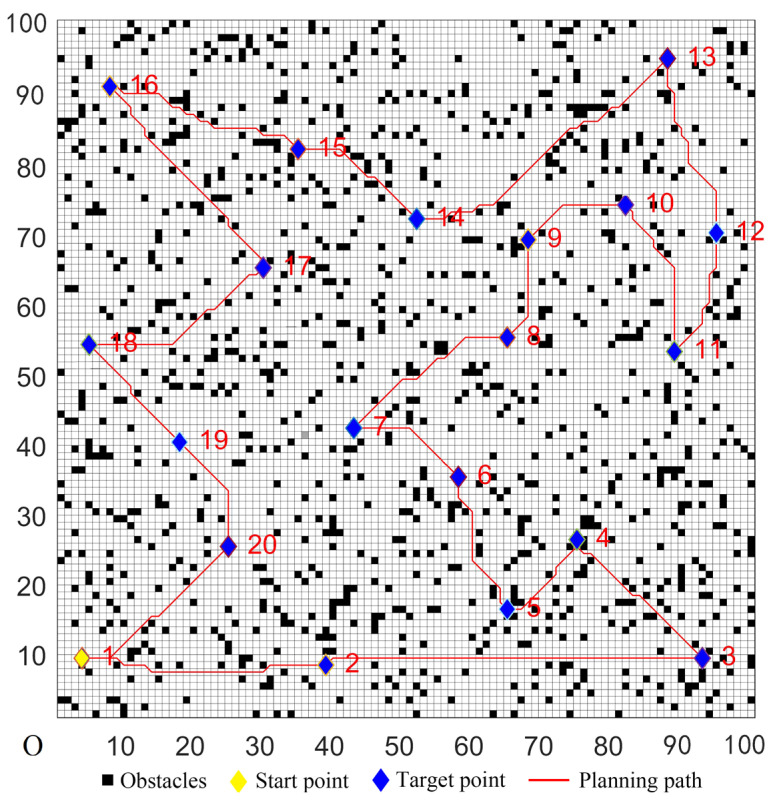
Multi-target path planning results of Algorithm 3 in complex environments.

**Figure 14 sensors-22-02429-f014:**
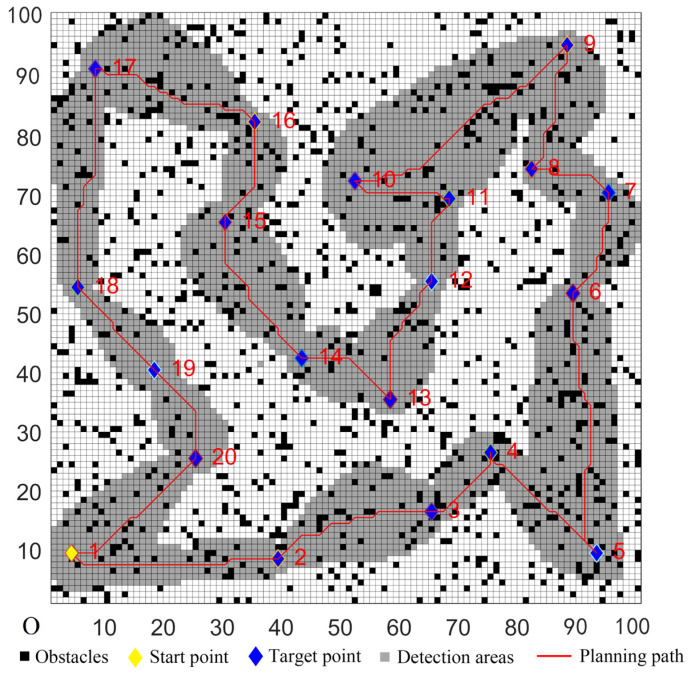
Multi-target path planning results of Algorithm 4 in complex environments.

**Figure 15 sensors-22-02429-f015:**
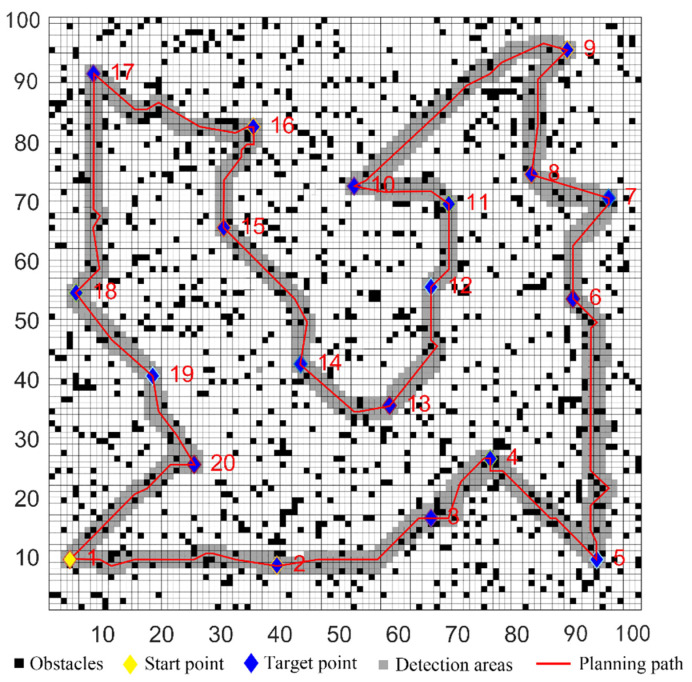
Multi-target path planning results of the proposed hybrid algorithm in complex environments.

**Figure 16 sensors-22-02429-f016:**
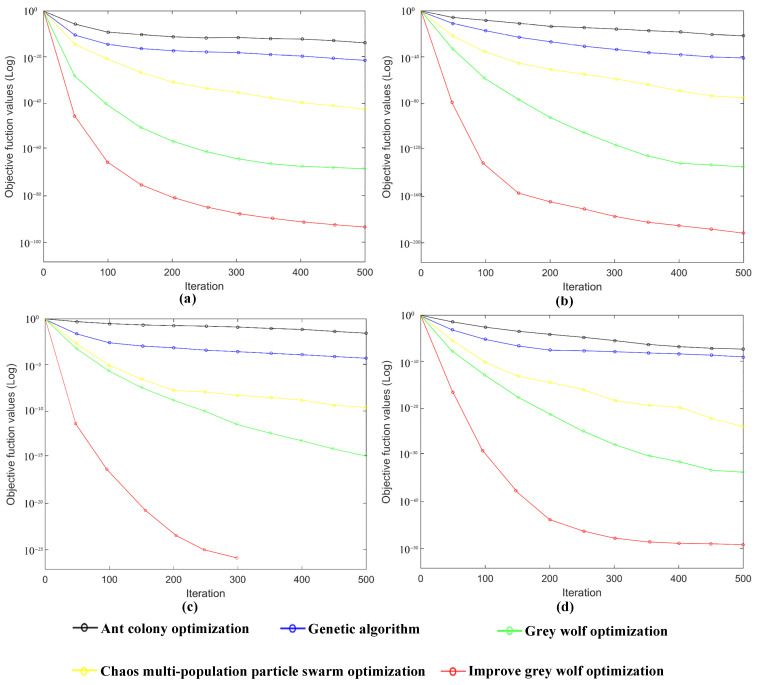
(**a**) Convergence curves of the five algorithms for *f_1_* function. (**b**) Convergence curves of the five algorithms for *f*_2_ function. (**c**) Convergence curves of the five algorithms for *f*_3_ function. (**d**) Convergence curves of the five algorithms for *f*_4_ function.

**Table 1 sensors-22-02429-t001:** Target Coordinates in Ordinary Environments.

Case	Target Coordinate
1	(4, 46), (8, 28), (6, 19), (14, 10), (17, 22), (26, 29), (29, 11), (41, 4), (39, 26), (37, 41)
2	(37, 26), (38, 3), (26, 9), (11, 8), (2, 18), (3, 24), (6, 27), (16, 28), (25, 32), (36, 30)
3	(17, 23), (31, 22), (8, 9), (11, 11), (19, 18), (5, 14), (38, 17), (26, 18), (14, 33), (35, 37)
4	(26, 24), (13, 31), (6, 9), (15, 7), (12, 28), (33, 24), (6, 4), (36, 33), (16, 37), (6, 13)

**Table 2 sensors-22-02429-t002:** Simulation Parameters of Algorithm 4 and Proposed Hybrid Algorithm.

Symbol	Definition	Numerical Value
*n_w_*	Number of grey wolves	20
*t* _max_	Maximum number of iterations	200
*μ* _1_ *, μ* _2_	Position adjustment factors	0.01, 0.1
*λ* _1_ *, λ* _2_	Speed adjustment factors	1, 0.1
*U*	Priority list	∅
*k* _s_	Initial value of *k*_m_	0
*rhs*(*s*)	Path cost of node *s*	∞
*rhs*(*s*_goal_)	Path cost of node *s*_goal_	0
*g*(*s*)	Actual path cost of node *s*	∞

**Table 3 sensors-22-02429-t003:** Statistical Results Analysis of the Five Algorithms in Ordinary Environments in Case1.

Performance Indicator	Statistics	Algorithm 1	Algorithm 2	Algorithm 3	Algorithm 4	Proposed Algorithm
Planning time (s)	Best	21.562	10.185	9.667	12.927	9.746
	Mean	30.112	10.671	10.470	13.823	10.356
	Worst	37.608	11.145	11.293	14.773	11.106
	Std. Dev.	5.134	0.626	1.015	1.035	0.483
	*t*-test	2.1233 × 10^−10^ (+)	0.779	4.7421 × 10^−5^ (+)	5.7973 × 10^−6^ (+)	---
Planning length (m)	Best	1677.354	1769.300	1695.650	1698.624	1669.643
	Mean	1693.280	1812.457	1728.564	1708.821	1678.002
	Worst	1720.697	1855.835	1760.541	1721.100	1691.8235
	Std. Dev.	12.171	30.153	15.851	7.511	5.981
	*t*-test	0.000067 (+)	8.5439 × 10^−6^ (+)	6.8328 × 10^−9^ (+)	5.8455 × 10^−8^ (+)	---
Number of inflection points	Best	54.000	53.000	48.000	38.000	35.000
	Mean	56.360	62.560	51.540	42.020	36.740
	Worst	59.000	79.000	56.000	44.000	38.000
	Std. Dev.	1.764	7.919	2.566	1.937	0.906
	*t*-test	4.7190 × 10^−13^ (+)	6.9541 × 10^−7^ (+)	2.7559 × 10^−10^ (+)	3.3352 × 10^−9^ (+)	---

**Table 4 sensors-22-02429-t004:** Performance Comparison of the Five Algorithms in Ordinary Environments in Four Cases.

Case	Performance Indicator	Algorithm 1	Algorithm 2	Algorithm 3	Algorithm 4	Proposed Algorithm
1	Planning time (s)	29.583	11.176	10.740	13.823	10.338
	Planning length (m)	1695.393	1816.817	1731.559	1708.821	1679.052
	Number of inflection points	56	61	52	39	35
2	Planning time (s)	25.235	10.861	9.198	11.043	8.977
	Planning length (m)	1468.860	1647.985	1502.314	1532.208	1443.670
	Number of inflection points	36	54	40	32	26
3	Planning time (s)	26.402	12.400	11.271	13.043	9.853
	Planning length (m)	1566.704	1691.964	1629.008	1630.361	1542.882
	Number of inflection points	51	70	49	42	34
4	Planning time (s)	21.743	8.671	8.231	11.562	8.174
	Planning length (m)	1230.362	1339.065	1276.521	1298.388	1210.675
	Number of inflection points	32	44	39	29	26

**Table 5 sensors-22-02429-t005:** Target Coordinates in Complex Environments.

Case	Target Coordinates
1	(4, 9), (39, 8), (65, 16), (75, 26), (93, 9), (89, 53), (95, 70), (82, 74), (88, 95), (52, 72), (68, 69), (65, 55), (58, 35), (43, 42), (30, 42), (35, 82), (8, 91), (5, 54), (18, 40), (25, 25)
2	(3, 19), (4, 38), (8, 28), (13, 10), (14, 75), (17, 22), (22, 32), (25, 50), (28, 11), (30, 30), (38, 25), (38, 51), (39, 5), (39, 64), (53, 76), (58, 40), (63, 60), (70, 13), (75, 77), (76, 42)
3	(17, 54), (56, 82), (21, 25), (26, 28), (31, 10), (30, 72), (36, 15), (36, 36), (40, 92), (45, 34), (98, 72), (56, 23), (53, 54), (49, 29), (60, 61), (67, 20), (69, 77), (74, 37), (23, 58), (93, 61)
4	(56, 3), (46, 34), (12, 67), (6, 28), (61, 45), (90, 12), (46, 87), (61, 63), (64, 72), (87, 34), (58, 52), (46, 43), (33, 51), (39, 39), (65, 61), (87, 70), (19, 87), (4, 35), (7, 5), (50, 94)

**Table 6 sensors-22-02429-t006:** Statistical Results Analysis of the Five Algorithms in Complex Environments.

Performance Indicator	Statistics	Algorithm 1	Algorithm 2	Algorithm 3	Algorithm 4	Proposed Algorithm
Planning time (s)	Best	69.319	22.216	21.037	29.200	20.742
	Mean	76.248	25.635	23.714	31.234	22.891
	Worst	82.667	27.531	24.311	35.334	23.921
	Std. Dev.	4.387	1.532	1.472	1.989	1.121
	*t*-test	5.1372 × 10^−25^ (+)	0.000084 (+)	0.000454 (+)	6.8413 × 10^−8^ (+)	---
Planning length (m)	Best	4945.233	5186.258	5166.254	4998.402	4804.200
	Mean	5034.751	5262.745	5219.564	5048.473	4841.964
	Worst	5141.769	5402.847	5287.889	5109.630	4897.646
	Std. Dev.	52.138	103.700	64.732	47.351	33.136
	*t*-test	1.8643 × 10^−8^ (+)	1.931 × 10^−7^ (+)	2.6843 × 10^−12^ (+)	4.2784 × 10^−6^ (+)	---
Numbers of inflection points	Best	103.000	136.000	106.000	92.000	74.000
	Mean	115.460	150.200	126.780	104.760	80.640
	Worst	130.000	184.000	154.000	117.000	93.000
	Std. Dev.	12.568	20.183	23.976	10.806	8.585
	*t*-test	3.3506 × 10^−19^ (+)	1.746 × 10^−11^ (+)	8.1961 × 10^−21^ (+)	7.1776 × 10^−15^ (+)	---

**Table 7 sensors-22-02429-t007:** Performance Comparison in Complex Environments for Four Cases.

Case	Performance Indicator	Algorithm 1	Algorithm 2	Algorithm 3	Algorithm 4	Proposed Algorithm
1	Planning time (s)	78.278	25.680	23.714	31.564	22.891
	Planning length (m)	5012.741	5292.691	5263.574	5054.457	4830.673
	Numbers of inflection points	107	150	106	104	79
2	Planning time (s)	66.327	24.010	23.207	25.347	22.355
	Planning length (m)	4051.769	4511.827	4309.645	4109.230	3989.236
	Numbers of inflection points	114	146	102	99	76
3	Planning time (s)	64.283	25.738	25.211	28.576	24.336
	Planning length (m)	4366.763	4685.202	4554.248	4568.221	4279.043
	Numbers of inflection points	99	133	110	101	84
4	Planning time (s)	92.273	28.472	27.393	36.371	26.284
	Planning length (m)	5264.147	5668.923	5554.954	5461.986	5093.817
	Numbers of inflection points	134	180	156	143	122

**Table 8 sensors-22-02429-t008:** Test Functions Used in Experiments.

Function Type	Function Name	Function Formula	Dimension	Search Range	*f* _min_
Unimodal function	Sphere	$f_{1}(x)=\sum_{i=1}^{n}x_{i}{}^{2}$	30	[−100, 100]	0
	Schwefel’s 2.21	$f_{2}(x)=\max_{i}\{\left|x_{i}\right|,1\leq i\leq n\}$	30	[−100, 100]	0
Multimodal function	Rastrigin	$f_{3}(x)=\sum_{i=1}^{n}\left[x_{i}{}^{2}-10\cos(2\pi x_{i})+10\right]$	30	[−5.12, 5.12]	0
	Alpine	$f_{4}(x)=\sum_{i=1}^{n}\left|x_{i}\sin(x_{i})+0.1x_{i}\right|$	30	[−10, 10]	0

**Table 9 sensors-22-02429-t009:** Experimental Results Obtained by the Five Algorithms for Four Test Functions.

Test Function	Statistics	Ant Colony Optimization	Genetic Algorithm	Chaos Multi-Population Particle Swarm Optimization	Grey Wolf Optimization	Proposed Algorithm
*f* _1_	Mean	4.62 × 10^−16^	5.01 × 10^−22^	2.58 × 10^−43^	1.21 × 10^−69^	3.23 × 10^−93^
	Std. Dev.	3.49 × 10^−16^	1.87 × 10^−22^	2.12 × 10^−43^	7.38 × 10^−69^	4.66 × 10^−93^
*f* _2_	Mean	9.28 × 10^−20^	9.31 × 10^−42^	9.35 × 10^−77^	3.66 × 10^−135^	7.57 × 10^−191^
	Std. Dev.	5.72 × 10^−20^	7.63 × 10^−42^	6.27 × 10^−77^	2.98 × 10^−135^	8.43 × 10^−191^
*f* _3_	Mean	1.24 × 10^−2^	5.78 × 10^−4^	6.95 × 10^−9^	1.71 × 10^−15^	0
	Std. Dev.	9.36 × 10^−2^	2.33 × 10^−4^	3.25 × 10^−9^	0.62 × 10^−15^	0
*f* _4_	Mean	5.32 × 10^−7^	4.23 × 10^−9^	1.09 × 10^−24^	9.54 × 10^−34^	1.95 × 10^−49^
	Std. Dev.	7.14 × 10^−7^	2.11 × 10^−9^	1.87 × 10^−24^	6.03 × 10^−34^	5.82 × 10^−49^

## Data Availability

Not applicable.

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
