# Peer review of "A Hybrid Multi-Target Path Planning Algorithm for Unmanned Cruise Ship in an Unknown Obstacle Environment"

_sensors, 2022, doi:10.3390/s22072429_

Round 1
Reviewer 1 Report
Reviewer’s comments
to the manuscript “A Hybrid Multi-target Path Planning Algorithm for Unmanned Cruise Ship in an Unknown Obstacle Environment" (Authors: Jiabin Yu, Guandong Liu, Jiping Xu, Zhiyao Zhao, Zhihao Chen, Meng Yang, Xiaoyi Wang, and Yuting Bai).
The article is devoted for the solving the problem of traversal multi-target path planning for an unmanned cruise ship in an unknown obstacle environment of lakes, this study proposes a hybrid multi-target path planning algorithm. To solve the problem, the authors proposed an improved algorithm based on the optimization of the gray wolf algorithm (GWO). Optimization of the algorithm consists in improving convergence and increasing the speed of operation. The algorithm was tested in numerical experiments and compared with other well-known algorithms.
There are some other points to correct or to make the information more exact:
Essential drawbacks.
Remark 1. Contradiction in lines 73-74 “the GWO algorithm has fast convergence speed, high solution accuracy, simple algorithm operation, only a few parameters to be set, and high robustness.” and lines 79-80 “However, the aforementioned GWO algorithms have slow convergence speed and high calculation cost.”
Remark 2.
Lines 146-180. “2.2. GWO Algorithm” Equations describing this algorithm are well-known, so there is no need to describe them in detail. It is necessary to denote vectors as vectors so as not to be confused with scalars.
Lines 181-213. 2.3. “D* Lite Algorithm”. A similar situation as in the case above.
Remark 3. Line 240. Equation 15. What is the meaning of the Sj?
Remark 4. Lines 249-251. The authors again write about the slow convergence of the algorithm, although the introduction indicated the opposite (see lines 73-74).
Remark 5. Line 261. Equation 16. There is no difference between a scalar and a vector (array). Need to reassign.
Remark 6. Line 280. Equation 19. The equation is not clear. Is it a minimization of the length of a vector, i.e., its Euclidean norm?
Remark 7. Line 295. Equation 20. What does P and Q mean? Why are they in the text of this article? It is necessary to link this with the mathematics of the article or simply remove the equation and give a link to the literature.
Remark 8. Lines 304-305. Are this purely empirical coefficients (µ1, µ2, λ1, λ2)? What is the mechanism of these coefficients?
Remark 9. Lines 314-315. Figure 2. What convergence factor is used here? There is a factor - Rate of convergence. Why not use it as a generally accepted characteristic of numerical methods? Is it possible to give some integral assessment?
Remark 10. Line 332. It is very difficult to read when the equations are in one place and their application in another. It would be necessary to restructure the article so that everything is reduced to one place.
Remark 11. Line 360. Equation 22. How is this equation obtained? Is this a purely empirical selection?
Remark 12. Line 365. Figure 3. The designations in the figure are very small. Nothing is visible.
Remark 13. Line 378. Figure 4. Unreadable figure. If you compare 2 algorithms, you need to combine Figure 3 and 4 and show the differences in one figure. Which, in principle, has already been done in Figure 5.
Remark 14. Line 432. Table 1. What are these coordinates mean and how are they related to the real map? What are the units of measurement?
Remark 15. Line 457. Table 2. k is an evaluation function, not an update variable. What does the numerical value of φ mean?
Remark 16. Line 497. Table 3. How is the time in seconds obtained? What is the speed of the ship? Is it constant or variable? Maybe some impersonal assessment is better than a second?
Remark 17. Line 497. Table 3. Why don't the authors provide t-test for their algorithm? If this really cannot be done, then why use this criterion to evaluate the effectiveness of your algorithm?
Remark 18. Line 497. Table 3. Meters are given for Planning path, but there are no units in the table of points and in the figures.
Remark 19. Line 499. Table 4. How the cases in Table 1, the usual conditions in Table 3 and Figures 6-10 are related. It is necessary to give information more structurally. Problem statement and result. These cases are not shown on the map and what is the difference between each of them is not clear.
Remark 20. Line 527. What is the function f1–f4? Why enter a designation without description?
Remark 21. Lines 536-544. Why are there gray areas in Figures 14–15, but not in the previous Figures 11–13?
Remark 22. Line 547-554. Figure 16–19. What is postponed along the y axis? In Figure 3, the authors had a convergence factor. Here OFV. In my opinion, there is confusion here.
Remark 23. Table 6. Why take the t-test criterion that the authors cannot calculate for their own algorithm?
Remark 24. Table 8. This is interesting, but clearly the table should not be here, but in the section describing algorithms with a specific indication of where which functions are used. And most importantly, since you write about them, why exactly these functions?
Remark 25. Line 569. Table 9. What are these functions (f1–f4) and why are they random?
Remark 26. Line 577. What other four algorithms?
Remark 27. Line 581. It is necessary to specify what is better and by how much.
Remark 28. Line 582. Here it is necessary to write what application, winnings, etc. the developed algorithm will give for people. It is necessary to show why this work was done at all, except for the personal interest of the authors.
Technical drawbacks.
Remark 1 Lines 189-213, 230-241, 258-283, 332-364, 369-373, 444-455. Font walks. Symbols and text are not on the same line.
Remark 2. Line 192. Equation 9. The minimum condition is written under the symbol min.
Remark 3. Line 332. It is very difficult to read when the equations are in one place and their application in another. It would be necessary to restructure the article so that everything is reduced to one place.
Remark 4. Line 364. µ is traditionally denotes the Rate of convergence. This is an unfortunate designation for a weight multiplier.
Remark 5. Line 577. Use past simple or present perfect.
Reviewer 2 Report
The work is devoted to the trajectories planning of an Unmanned Cruise Ship in an Unknown Obstacle Environment. In the article a hybrid algorithm based on the improved GWO-D*Lite algorithm is proposed. First, the multi-target planning problem is transformed into a TSP, and the improved GWO algorithm is used to plan a multi-target cruise sequence. Second, based on the obtained sequence, the improved D*Lite algorithm is used for path planning between every two target points.
The work has a large number of comments.
- Lines 108, 109 have different spelling of D*Lite algorithms.
- Equations (1)-(4) the constant tmin is not explained.
- Line 167 apparently we are talking about the components of the vector that are evenly distributed? Or is it a vector module?
- Equations (5)-(8), discrete time is introduced in the work? What does t+1 mean?
- Mathematical formulas are written extremely carelessly in the general text they come out of the lines and have different font sizes, in individual formulas, not all variables are explained and correspond to the presentation.
- Formula (13) is part of some algorithm, as it is given, it does not make sense.
- Formulas (17)-(18) the size of the matrices should be transferred to the text when defining them.
- Formula (19) it is not known by which variables the minimum is taken.
- Formula (20) What does x mean?
Reviewer 3 Report
In this article, authors propose a hybrid multi-target path planning algorithm for unmanned cruise ship. However, I comment on some aspects to improve the quality of the manuscript:
-Authors use incorrectly acronyms. The correct form of acronyms is to write the meaning of the acronym with the capital letter that represents the acronym. This error must be corrected in all the acronyms that the authors have written.
-The authors have not written a Related Works Section.
-The authors must improve all the Algorithms written in the manuscript, as a suggestion for the authors, they must review from other articles how they use the algorithms.
-Algorithms have not been cited.
-Where did the authors obtain the data to perform the simulations?
-What is the reason that the authors have only performed 10 min for the simulation?
-Which are the 5 algorithms mentioned in line 459, must be specified?
-When citing an object such as Figure, Algorithm, Section, Equation, Table, it must be completely written and not abbreviated.
-The proposed algorithm up to what level of complexity can give a solution?
-In Figure 16, the legend must mention what algorithm it is and not with acronyms.
-Avoid the use of phrasal verbs in a scientific article.
-Improve the quality of English.
-The authors must improve the conclusions.
Round 2
Reviewer 1 Report
Technical drawbacks.
Remark 1. Figure 1 is not centered relative to the article page.
Remark 2. Line 319. Equation 6. The font differs from the previous equations.
Remark 3. The figures 3 and 4 use a different font for the text. Should be the same.
Reviewer 2 Report
Unfortunately, the mathematical description of the work has deteriorated significantly.
In formula (1), the mathematical object located under the module is not clear. There is a product of vectors, what is this product? If scalar, then a vector is subtracted from the scalar, which is erroneous.
In formulas (2) and (14) there is confusion again, the right part is a scalar, and the left part is a vector. And so on in the text, formulas (3)-(5) contain similar errors.
Remark 5 has not been fully taken into account. Mathematical formulas have different font sizes.
Remark 8 is not taken into account. In formula (8) it is not known by which variables the minimum is taken. If the optimization parameter is strategy I, then where did it come from in the right part (8).
Formula (13) is better written as f=\min_{j}\sum_{m=1}^{d}s_{j,d}.
Formulas (11), (12) have different index designations for matrix elements.
A vector is usually considered a column vector, and the transpose operation gives a row vector.
Line 588 again contains not an equation, but a part of the algorithm. You should write k_m:=k_m+h.
Reviewer 3 Report
Thanks to the authors for performing the changes suggested by the reviewers. However, there are details that need to be improved before publication:
-Authors must review the correct format for the Journal, because there are some errors such as insufficient space between the objects and the text of the manuscript, as occurs in line 681.
-Eliminate the vertical line on the left side of the manuscript, as well as the gray box that appears on some pages that corresponds to the Track Change of Microsoft Word.
-Improve the quality of scientific English, eliminating the Phrasal Verbs that the authors have used.
Round 3
Reviewer 2 Report
No further comments.